# Recombinant Oxidase from *Armillaria tabescens* as a Potential Tool for Aflatoxin B1 Degradation in Contaminated Cereal Grain

**DOI:** 10.3390/toxins15120678

**Published:** 2023-11-30

**Authors:** Igor Sinelnikov, Oleg Mikityuk, Larisa Shcherbakova, Tatyana Nazarova, Yury Denisenko, Alexandra Rozhkova, Natalia Statsyuk, Ivan Zorov

**Affiliations:** 1Federal Research Centre “Fundamentals of Biotechnology” of the Russian Academy of Sciences, 119071 Moscow, Russia; denisenkoyura@mail.ru (Y.D.); a.rojkova@fbras.ru (A.R.); inzorov@mail.ru (I.Z.); 2All-Russian Research Institute of Phytopathology, Bolshie Vyazemy, 143050 Moscow, Russia; mod-39@list.ru (O.M.); nataafg@gmail.com (N.S.)

**Keywords:** aflatoxin B1, enzymatic toxin degradation, recombinant enzymes, grain decontamination

## Abstract

Forage grain contamination with aflatoxin B1 (AFB1) is a global problem, so its detoxification with the aim of providing feed safety and cost-efficiency is still a relevant issue. AFB1 degradation by microbial enzymes is considered to be a promising detoxification approach. In this study, we modified an previously developed *Pichia pastoris* GS115 expression system using a chimeric signal peptide to obtain a new recombinant producer of extracellular AFB1 oxidase (AFO) from *Armillaria tabescens* (the yield of 0.3 g/L), purified AFO, and selected optimal conditions for AFO-induced AFB1 removal from model solutions. After a 72 h exposure of the AFB1 solution to AFO at pH 6.0 and 30 °C, 80% of the AFB1 was degraded. Treatments with AFO also significantly reduced the AFB1 content in wheat and corn grain inoculated with *Aspergillus flavus*. In grain samples contaminated with several dozen micrograms of AFB1 per kg, a 48 h exposure to AFO resulted in at least double the reduction in grain contamination compared to the control, while the same treatment of more significantly (~mg/kg) AFB1-polluted samples reduced their contamination by ~40%. These findings prove the potential of the tested AFO for cereal grain decontamination and suggest that additional studies to stabilize AFO and improve its AFB1-degrading efficacy are required.

## 1. Introduction

In spite of the different approaches developed for the prevention of and reduction in food and feed pollution with aflatoxin B1 (AFB1) [1,2,3,4,5,6], this widespread and very carcinogenic mycotoxin [7,8] continues to be a major concern as a contaminant of various feeds, including wheat, corn and other cereals [9,10]. Cereal grain and its primary processing products constitute the bulk of feed rations for many farm animals [11], especially monogastric ones, including poultry. Wheat and corn are the main components of the poultry feed diet in European countries and the United States, respectively [12,13]. At the same time, according to the Food and Agricultural Organization (FAO) and other sources, 25% of the total production of the world’s food and feed crops, including wheat, corn and other cereals, are contaminated with mycotoxins [14,15]. In poultry farming, problems associated with the grain contamination with AFB1 are exacerbated by a high sensitivity of poultry, especially turkeys and ducks (LD_50_ 1.4–3.2 and 0.3–0.6 mg/kg of body weight, respectively) and, to a lesser extent, quails and chickens, to AFB1 [3,16,17,18], as well as by the fact that the feed cost can exceed 70% of the poultry meat cost [19]. Therefore, decontamination of cereal grain with the aim of providing the feed safety and cost efficiency still remains an urgent challenge. 

In this regard, enzymatic degradation of AFB1 based on the ability of some fungi and bacteria [20,21,22,23] to synthesize various enzymes (including oxidoreductases) transforming this toxin into non- or less-toxic compounds is considered to be one of the most promising approaches [24,25,26,27,28,29,30]. The use of cell-free preparations containing such enzymes could make it possible to avoid the problems which may occur during the use of their producers (worsening of organoleptic properties of treated products, reduction in their nutritional value, etc.). Moreover, enzymatic preparations are more technologically suitable for the feed treatment [31] and, unlike similar preparations for the food industry, do not require a high level of purification of a target product.

Some xylotrophic basidiomycetes from the genera *Pleurotus*, *Phanerochaete*, *Armillaria*, *Peniophora* and *Trametes* [20,32,33,34] are well known as the sources of microbial oxidoreductases able to degrade or detoxify aflatoxins. For example, manganese-dependent peroxidase from *Phanerochaete sordida* [35], certain laccases from *Trametes versicolor*, *Pleurotus ostreatus,* and *Peniophora* sp., as well as recombinant laccase secreted by *A. niger* [20] are shown to actively degrade AFB1, AFB2, and G-type aflatoxins [27,36], and may be considered as possible tools for the decontamination of agroproducts. However, some limitations may hinder the development of decontaminating preparations based on these enzymes. For example, *P. sordida* was reported to be able to infect humans [37]. Laccases are well-known enzymes catabolizing many compounds including lignin, and are considered for use in wood processing and the textile industry [38] because of the broader specificity of this group of enzymes compared to other aflatoxin-degrading enzymes and their activity at extreme pH values and temperatures [39,40], which are not typical for feed processing. Moreover, in spite of the successful results demonstrated in some studies [27,36,40], the heterologous expression of laccases is rather difficult due to their complex copper-dependent active site [38,40] that results in a rapid proteolytic cleavage of these enzymes during their post-translational modification. Therefore, studies aimed at searching for new promising enzymes for feed decontamination still remain relevant. 

Some studies suggest that one of such tools may be an intracellular enzyme possessing oxydase activity, which was isolated from the *Armillaria tabescens* mycelium as a component of a multi-enzyme complex of this edible fungus [34]; the authors of the study named it aflatoxin-detoxifizyme. This enzyme was later identified as a new AFB1 oxidase (AFO) able to catalyze the opening and the further hydrolysis of a bisfuran ring [41] (a structure directly associated with AFB1 toxicity [42,43]). An unsaturated C8–C9 bond was determined to be a putative reactive AFO site in the AFB1 molecule [43]. A further study showed that AFO represented a 76 kDa monomeric protein, which significantly differed from aforementioned oxidoreductases; the contact of this protein with the toxin caused a significant reduction in AFB1 toxicity and mutagenicity [34]. This enzyme possesses a low K_m_ value (0.334 µmol/L) for AFB1 that explains its high affinity to AFB1 [41,43]. In addition, the highly selective detoxifying properties of AFO were shown not only for AFB1, but also for sterigmatocystin and versicolorin A, the toxic AFB1 precursors synthesized by many *Aspergillus* species [42,43,44]. Moreover, the hydrogen peroxide production that occurs during the enzymatic reaction results in the detoxification of AFB1-epoxide, a highly toxic derivative formed at the first stage of AFB1 oxidation. These data evidence the good prospects for the development of detoxifying preparations based on AFB1 oxidase (AFO) from *A. tabescens*. However, their development requires an available technology of AFO production and secretion that can be achieved via the use of a system for a heterologous expression of a recombinant enzyme. A few earlier reports [45,46,47] as well as our preliminary study [48] on recombinant AFO production indicated the possibility of using transformed *Pichia pastoris* strains as the appropriate heterologous producers of this enzyme. The authors of these in vitro studies demonstrated the ability of the obtained recombinant AFOs to degrade AFB1 and determined some kinetic characteristics and the target activity level of the enzymes at different pH levels and temperatures [45,46,47,48]. To produce the recombinant AFO in the *P. pastoris* GS115 strain in our earlier study [48], we optimized and used a previously developed expression system with a modified integration vector intended to increase the copy number variation of heterologous genes in yeast cell chromosomes. As a result, the most productive ADTZ-14 clone, which removed 50% of AFB1 after a 72 h incubation of the toxin in its culture liquid, was selected among 54 obtained transformants. 

Continuing our investigation, in the current study, we modified the previously used *P. pastoris* GS115 expression system using a new chimeric signal peptide and generated a new highly productive recombinant strain to obtain extracellular heterologous AFO in an amount sufficient for forage grain decontamination. We also homogenously isolated the recombinant enzyme and selected optimal conditions by varying the pH and temperature of incubation medium to achieve the best target effect (AFB1 removal) in model solutions. In addition, we showed that the AFO treatment of wheat and corn grain, contaminated with AFB1 via inoculation with *A. flavus*, resulted under selected conditions in a significant reduction in the AFB1 content in grain. 

## 2. Results

### 2.1. Recombinant Aflatoxin B1 Oxidase Obtained Using a Pichia Pastoris Expression System

The synthetic *afo* gene encoding aflatoxin-degrading oxidase from *Armillaria tabescens* (GenBank AY941095.1) was synthesized by Eurogen Ltd. (Moscow, Russia) according to the codon usage of *Pichia pastoris* (syn. *Komagataella phaffii*). The sequence was cloned by PCR using specially designed oligonucleotides; the size of the resulting PCR product was 2088 bp.

To construct a plasmid for the *afo* integration, a modified pPIG-1 vector was used. The pPIGa-1 vector contained an alcohol oxidase I (AOXI) promoter instead of a GAP promoter. In contrast to the previous study [48], the *afo* sequence was expanded with a chimeric signal peptide, which included a pro-region of the α-factor and a synthetic signal peptide (MKILSALLLLFTLAFA, https://pubmed.ncbi.nlm.nih.gov/25218497/, accessed on 22 November 2023) facilitating efficient co-translational transport into the endoplasmic reticulum. After transformation and selection (see Section 4.1), we obtained a recombinant *P. pastoris_afo* strain characterized by a 0.3 g/L yield of the target aflatoxin B1 oxidase (AFO) and reduced degradation of this enzyme.

The supernatant of the culture liquid (50 mL) was applied to a 15 mL Ni-NTA Sepharose column, resulting in the isolation of homogeneous AFO via chromatographic separation. The AFO fraction had a concentration of 1.3 mg/mL. The size and purity of the enzyme evaluated by means of denaturing gel electrophoresis (Figure 1) were 72 kDA and ~95%, respectively.

### 2.2. Enzymatic AFB1 Degradation in Model Solutions under Different Conditions

In order to confirm that the obtained recombinant AFO retained its target functionality and to select conditions promoting its toxin-degrading activity, the experiments on the AFB1 removal from the model solutions were arranged. In the course of these experiments, AFB1 degradation was examined through a 5-day toxin incubation in the enzyme-containing buffers at different pH and temperatures followed by a reverse-phase HPLC to determine a residual AFB1 content. The dynamics of the AFB1 removal from the model solutions significantly depended on both studied factors. The pH and temperature ranges used in these experiments were chosen on the basis of previous results obtained for AFO produced by *A. tabescens* [34,41] or transformed *P. pastoris* G115 clones [47,48], respectively.

In general, AFO showed the target activity at pH levels varying between 5.0 and 7.5, while more acidic (pH 4.0) or alkaline (pH 8.0 and 9.0) conditions resulted in an almost complete inactivation of the enzyme. The most effective AFB1 removal was observed in the case of a 3-day exposure of the model solution to the enzyme at pH 6.0 (Figure 2). In this case, less than half of the added toxin was detected in a 20 mM phosphate buffer after a 24 h exposure. After a 48 h co-incubation, the average AFB1 content reduced to 30% of the initial level; finally, after a 72 h exposure, the residual AFB1 content in the model solution was reduced to 20% of the initial content. No additional significant removal of AFB1 was observed in the course of the further incubation, and the toxin degradation level remained at 80–81% for the next two days until the completion of the experiment (Figure 2); even a 2.5-fold increase in the AFO concentration did not enhance or prolong the target effect. The same trend for the termination of the AFB1 removal from model solutions after 72 h of incubation was revealed for all other tested pH values (Figure 2). Based on these data, we further investigated the effect of three different temperatures (15, 30 and 50 °C) on the enzymatic toxin degradation at optimal pH (6.0) and recorded the 72 h toxin removal dynamics (Figure 3).

In our study, the most rapid decrease in the AFB1 content caused by the AFO treatment was observed at 30 °C. In this case, after 24, 48, and 72 h of incubation, ~50, 66 and 80% of the toxin was degraded, respectively. Incubation at 15 °C significantly slowed down the degradation process, resulting in a removal of only ~40% of AFB1 within a 48 h incubation. A temperature increase to 50 °C resulted in enzyme inactivation (Figure 3).

### 2.3. Enzymatic Decontamination of Cereal Grain Inoculated with a Toxigenic Aspergillus flaus 

To examine if the obtained recombinant oxidase was able to reduce aflatoxin contamination of a polluted grain, a toxigenic *A. flavus* strain was grown on autoclaved wheat or corn grain for 3 or 7 days. Then, inoculated grain samples were treated with 20 mM phosphate buffer (control) or AFO, dissolved in the same buffer, for 24, 48 and 72 h under the aforementioned conditions with the further extraction and quantification of residual AFB1. The fungus actively developed on a grain; by the third day of cultivation, 47.0 ± 7.0 and 33.0 ± 5.8 ng of AFB1 per gram of grain was accumulated in the wheat and corn samples, respectively. By the end of cultivation, AFB1 concentrations in non-treated inoculated grain samples reached the microgram level (1.08 ± 0.20 and 0.99 ± 0.14 µg/g for wheat and corn grain, respectively). Thus, the average content of AFB1 in the analyzed samples calculated per kg of grain was ~30–50 and 990–1080 µg at the beginning and at the end of a 10-day experiment, respectively.

In contrast, a significantly lower amount of the toxin was determined in AFO-treated grain (Figure 4). For the majority of wheat and corn samples, which contained a few dozen micrograms of AFB1 per kg, a 48 h exposure to AFO reduced the grain contamination level to at least twice the extent compared to the control (Figure 4a), while the same treatment of more heavily contaminated samples reduced the AFB1 content by 40% on average (Figure 4b). 

Regardless of the initial amount of AFB1, a general downward trend in the toxin content in contaminated wheat and corn grain was also observed after a 3-day treatment; in the case of a low-contamination corn grain, the achieved toxin degradation level was 61.2% (Figure 4a). However, the difference between the results of the 48 and 72 h treatments should be considered rather insignificant because of the high standard deviation values. A longer exposure of inoculated grain to AFO did not enhance the decontamination effect. Nevertheless, for some samples analyzed 3 days after inoculation, almost a 3-fold decontamination was reached in one of the performed experiments with a 72 h exposure to AFO. As a result, the content of residual AFB1 was reduced to 16.8 and 10.6 µg/kg in wheat and corn grain, respectively (Figure 4a). The obtained data indicate that the tested AFO expressed in yeast cells was able to affect the toxin not only in model solutions but also in a contaminated cereal grain.

## 3. Discussion

To date, a wide range of recombinant enzymes able to attack mycotoxins have been produced using prokaryotic or eukaryotic expression systems [20,45,47,49,50,51,52]; for some of them, their decontaminating activity has been experimentally confirmed on grain and other food/feed products [45,49,51]. For example, a high decontamination potential was recently demonstrated for one recombinant AFO (Arm-ADTZ) in experiments on the enzymatic treatment of mold on corn (56.48% degradation within 24 h) and grain by-products obtained in the course of the distillation process in ethanol production [45]. 

AFO from *A. tabescens* seems to belong to the most studied AFB1-detoxifying enzymes including recombinant ones. Its crystal structure and the nucleotide sequence of a gene encoding a full-length protein have been determined [53], the safety of the resulting AFB1 degradation products has been confirmed [27], an additional dipeptidyl peptidase activity has been revealed [53], and at least three functional enzymes heterologously expressed in *P. pastoris* G115 have been reported [47,48]. Some kinetic characteristics, such as the K_m_ value, the specific activity, and the half-life period at 30–50 °C were described for one of these recombinant AFOs [45,47], as well as for an AFO isolated from *A. tabescens* [34,41]. In our study, the use of a new signaling peptide promoting effective AFO secretion allowed us to prevent a partial protein degradation observed in the previous experiments with the AFO-containing cultural liquid of transformed yeast cells [48] and to provide a sufficient production of the extracellular enzyme possessing the target activity. 

In the performed model experiments, the target AFO activity was revealed within the range of acidic and alkaline pH values (5.0–7.5 with the optimum at pH 6.0) used in the feed industry. AFO exposure to different temperatures indicated that the target activity was the most stable at 30 °C (temperature favorable for the toxin production). Such characteristics of the pH dependence as well as the revealed optimum temperature coincide with the data obtained for the parental enzymes from *A. tabescence* [34] and *Trametes versicolor* [49], as well as with results reported for another recombinant aflatoxin oxidase expressed in *P. pastoris* G115 [45,47]. Under optimal conditions, AFO rapidly degraded 80% of AFB after 72 h of incubation, while only 50% of AFB1 was in vitro degraded after its 72 h co-incubation with the enzyme under the optimal conditions reported in our earlier study [48]. Since an increase in the enzyme concentration did not result in additional toxin removal during its co-incubation with AFB1 for 96 or 120 h, the lack of any additional decontamination effect after a 72 h exposure was related to the enzyme’s destabilization rather than to the substrate substitution. This suggestion was confirmed by the significantly lower percentage of AFB1 degradation (max. 30.5%) in one of the 48 h incubation tests performed at the preliminary stage of our investigation; in this case, a prepared AFO solution was stored overnight prior to use. The addition of BaCl_2_ to the reaction medium [34] used in our experiments did not improve the AFO stabilization and did not provide the prolongation of its degradation effect.

A worldwide monitoring of feed contamination with AFB1 suggests that cereal grain has been affected by toxigenic Aspergilli (especially during storage) much more often in recent years [54,55,56]. Currently, the AFB1 limit in feeds in the European Union has been set by the FAO at the level of 5 µg/kg; the United States’ regulations on grain safety [10] and the Technical Regulations of the Customs Union (CU) restrict this level to 20 µg/kg (TR CU 015/2011). Because of the global warming-induced migration of *Aspergillus* fungi from tropical and subtropical regions, the incidence of finding cereal grain samples containing AFB1 and the level of the contamination tend to increase in zones with a temperate climate [56,57]; moreover, the further increase in the contamination risk is predicted [58]. Depending on the moisture level, *A. flavus* is able to grow on stored grain within the temperature range of 12–48 °C [59], with the most intense AFB1 production at 25–35 °C [60]. The AFB1 content in contaminated samples from various countries significantly varies. For example, the contamination of wheat and corn grain at the levels exceeding 30 µg/kg is often reported [10,61,62,63], and sometimes the maximum revealed AFB1 content can exceed 4 mg/kg [64]. Based on this information, we tested the obtained recombinant AFO on grain samples with the AFB1 content ranging from 33 to 1080 µg/kg and demonstrated that the enzyme can be effective even in the case of extremely high grain contamination. Despite the fact that the residual AFB1 content in wheat and grain samples, initially containing milligrams of AFB1 and exposed to AFO for 72 h, still significantly exceeded international and national maximum allowable limits, the similar treatment of less contaminated grain with the enzyme preparation resulted in a reduction in the residual AFB1 content below 20 µg/kg (the maximum allowable limit for CU countries and US). It is possible to suppose that a 2- or 3-day AFO application on non-heavily polluted samples may be more efficient and provide sufficient grain detoxification during this time period. Thus, we succeeded in obtaining a recombinant AFB1-degrading oxidase with promising properties of a potential grain decontamination agent. At the same time, the relatively low stability of the enzyme manifested via the loss of its target activity after 72- or 48 h incubation in model solutions or contaminated grain, respectively, constrains the prospects of its practical application at this stage and suggests that additional investigations are needed to improve the enzyme and to understand more about its detoxifying potential. In this regard, we will try to stabilize the obtained AFO via protein engineering methods in our further studies, and to test AFO activity on a contaminated grain of other cereals and crops characterized by a high risk of the AFB1 contamination. 

According to one of the current concepts [27], a transition to a large-scale production of detoxifying enzymatic and microbial preparations is only a matter of time, so, along with other studies, our study can contribute to the advancement of this eco-friendly approach to improve feed safety.

## 4. Materials and Methods

### 4.1. Generation of a Yeast Strain Producing Recombinant AFO

A methylotrophic yeast strain *Pichia pastoris* GS115 (Thermo Fisher Scientific Inc., Waltman, MA, USA) was used to express the *afo* gene encoding aflatoxin-detoxiphysim (AFO). 

Amplification of the *afo* gene was performed via routine PCR. The sequences for HindIII and NotI restriction endonucleases were added to the corresponding primers for the amplification of the *afo* gene. The corresponding primers were designed as follows: 

Forward: 5′-gaagcttcttctATGGCTATGGCTACTACTACAACTG-3′ (Hind*III*); 

Reverse: 5’-cgcggggccgcTTACAATCTTCTCTCTCTC-3´ (Not*I*). 

The expression plasmid pPIGA-afo was developed using the earlier described pPIG-I vector [48].

Both the pPIG-I vector and the amplification product were double digested by HindIII and NotI restriction endonucleases (New England Biolabs, Ipswich, MA, USA) in accordance with the manufacturer’s recommendations. The treated fragments were ligated with T4 DNA ligase (Eurogen Ltd., Moscow, Russia), and the ligated mixture (2 μL) was transformed into *E. coli* XL1-blue cells. Transformants were selected on the LB medium containing 2% agar supplemented with ampicillin (100 μg/mL, Sigma-Aldrich, St. Louis, MO, USA). To isolate plasmid DNA, *Escherichia coli* XL1-Blue strain (Agilent, Santa Clara, CA, USA) was grown at 37 °C on LB medium of the following composition (g/L): tryptone, 10; yeast extract, 5; NaCl, 5; pH 7.2–7.5). A pPIGA-afo plasmid was isolated from ampicillin-resistant transformants using a Plasmid Mini-prep kit (Eurogen Ltd., Moscow, Russia) and confirmed by means of PCR amplification, identification by double-enzyme digestion, and gene sequencing [65].

The pPIGA-afo plasmid (Appendix A) was linearized by ApaI restriction endonuclease (New England Biolabs, Ipswich, MA, USA) and integrated into *P. pastoris* GS115 by electroporation [66]. The selection of transformants was carried out on YPD medium with 1% agar supplemented by zeocin (200 μg/mL, Thermo Fisher Scientific Inc., Waltman, MA, USA). Total DNA samples were isolated from antibiotic-resistant colonies [67] and the presence of the *afo* insertion was verified by PCR followed by gene sequencing [65].

### 4.2. Recombinant AFO Production in Yeast Cells

The cultivation process included two stages: (1) biomass accumulation (24 h) and (2) protein synthesis induction by methanol. 

Fermentation was carried out in a 1.5 L Sartorius A plus bioreactor (Sartorius, Gottingen, Germany) containing 1 L of fermentation medium of the following composition: glycerol, 88 g/L; KH_2_PO_4_, 18.8 g/L; (NH4)_2_SO_4_, 31.4 g/L; MgSO_4_ × 7H_2_O, 9.2 g/L; CaCl_2_, 0.7 g/L; biotin, 0.4 mg/L; Sofaxil defoamer, 1 g/L. After autoclaving, the medium was supplemented with 4 mL of PTM1 trace element solution [68]. During fermentation, the temperature of cultivation and a medium pH were maintained at the same level (30 °C and 5.5, respectively). The dissolved oxygen content (DO) was maintained within the range of 40–70% by adjusting the stirring rate within the range from 200 to 500 rpm. 

A recombinant *P. pastoris* GS115-afo strain was grown in a 750 mL flask containing 100 mL of YPG medium. The strain was cultivated in an incubator shaker for 18–24 h at 30 °C and 250 rpm until reaching the optical density OD_600_ = 10; then, the whole volume of the culture was aseptically inoculated into a fermentation medium. After 16 h of growth on glycerol up to OD_600_ = 180–200, a periodic induction phase was started with the addition of 100% methanol with a feed rate of 1.5 mL/L/h. After 120 h of fermentation, cultural liquid was centrifuged for 20 min at 4000 rpm using an Avanti JXN-26 centrifuge (Beckman Coulter, Brea, CA, USA) and purified as described below.

### 4.3. AFO Purification Procedure 

Immobilized metal affinity chromatography (IMAC) was used for AFO purification. The centrifuged cultural liquid was applied onto a 15 mL Ni-NTA Sepharose excel column (Cytiva, Little Chalfont, UK) with a flow rate of 1.5 mL/min. The column was equilibrated with the equilibration buffer (20 mM potassium phosphate buffer (pH 7.4) plus 500 mM NaCl). To remove nonspecific bound proteins, the column was washed with a washing buffer (20 mM potassium phosphate buffer (pH 7.4), 500 mM NaCl, 20 mM imidazole) with a flow rate of 2 mL/min. The one-step elution was carried out using the elution buffer (20 mM potassium phosphate buffer (pH 7.4), 500 mM NaCl, 400 mM imidazole) with a flow rate of 2 mL/min. The quality of the AFO separation was evaluated using SDS-PAGE (Bio-RAD, Hercules, CA, USA). Homogeneous enzyme samples were dialyzed against double distilled water at 4 °C, freeze-dried using a SP VirTis Freeze Dryer (SP Scientific, Warminster, PA, USA), and stored at −20 °C until use.

### 4.4. AFB1 Degradation by AFO in Buffer Solutions 

To test the AFO’s ability to remove AFB1 from model solutions, freeze-dried samples of purified AFO were dissolved in 20 mM Na-acetate buffer (pH 4.0 or 5.0), or 20 mM Na-phosphate buffer (pH varied within 6.0–8.0), or 20 mM Tris-HCl buffer (pH 9.0). These solutions were sterilized by means of filtration through a 0.22 µm Millipore filter and supplemented with AFB1 (Sigma-Aldrich, St. Louis, MO, USA) dissolved in a minimum volume of methanol up to a final concentration of 0.25 µg/mL. The corresponding buffers supplemented with the toxin at the same concentration were used as controls. The model solutions were incubated for 120 h at 27 °C. During incubation, 50 µL aliquots were sampled every 24 h, diluted with the mobile phase (see Section 4.8) up to a final volume of 1 mL, and analyzed via HPLC as described below. After the selection of the buffer and optimal pH for the target AFO activity, the effect of different incubation temperatures (15, 30, and 50 °C) on the enzymatic degradation process was evaluated after 24, 48, and 72 h of incubation under selected conditions. The experiments were arranged into at least three biological replications.

### 4.5. AFB1 Degradation by AFO in Artificially Inoculated Grain

#### 4.5.1. Grain Inoculation with Aspergillus flavus 

A toxigenic *A. flavus* strain AF24 earlier isolated from peanut and stored at the laboratory collection as a stock culture was recovered and grown on potato dextrose agar. To prepare the inoculum, AF24 was sub-cultured on the same medium in a 9 cm Petri plate at 25 °C. When the sporulating aerial mycelium covered the whole surface of the medium, spores were collected by flooding the plate with sterile distilled water (SDW) to obtain their suspension.

The experiment was arranged in three biological replications. Samples of wheat grain or cracked corn kernels (50 g) were placed in 250 mL Erlenmeyer flasks (9 flasks per each grain type per one experiment), wetted with distilled water (20 mL per flask) and autoclaved for 1 h at 120 °C. Afterward, 2 mL of a spore suspension containing 10^6^ spores per mL of SDW was added into each flask. The flasks were shaken for a minute to thoroughly mix their content and evenly distribute the inoculum throughout the grain, placed into a thermostatic chamber, and incubated at 30 °C in the dark. Three days after inoculation, some of the flasks (9 flasks per each type of inoculated grain) were used for decontamination experiments, while the remaining flasks were left in the chamber for the next 4 days. Upon the end of the 3- or 7-day cultivation, the flasks were left overnight at a temperature below 12 °C to stop fungal growth and to prevent additional toxin production [59,69] during the subsequent treatment with enzyme at 30 °C. 

Flasks containing wheat or corn grain without *A. flavus* spores were further used as non-inoculated (negative) controls.

#### 4.5.2. Treatment of Inoculated Grain with AFO

A freeze-dried AFO preparation obtained by means of fermentation and purified as described above was used in grain decontamination experiments. On the day of treatment, a portion of the enzyme preparation was dissolved in 20 mM phosphate buffer (pH 6.0) to a final concentration of 50 mg/mL. Then, 100 mL of this solution was added into each flask and gently mixed with grain. In parallel, the same volume of the buffer was added into other flasks with the inoculated grain as inoculated (positive) controls. Thereafter, flasks were incubated at 30 °C with slow shaking for 48 or 72 h in the dark.

### 4.6. Isolation of Residual AFB1 from Cereal Grain after Enzymatic Decontamination

Upon completion of the incubation process, the flasks were supplemented with acetonitrile up to a final concentration of 80%. The grain was extracted for 1 h at room temperature and intensive shaking (100 rpm). Aliquots of the filtered acetonitrile extract equivalent to 5 g of grain were evaporated using a rotary evaporator up to the aqueous phase, which was further supplemented with a saturated NaCl solution up to the total volume of 20 mL. After the pH adjustment to 3.0, the solution was added to the equal volume of water-saturated hexane followed by the aqueous phase separation via extraction with dichloromethane. The final AFB1-containing extract was passed through a layer of anhydrous Na_2_SO_4_ and dried on a rotary evaporator. The residue was dissolved in a minimal volume of a mobile phase consisting of 0.3% H_3_PO_4_ and acetonitrile (1:1), filtered through a membrane with the pore size of 0.45 μm, and analyzed by means of reverse-phase HPLC. If necessary, the final extract was additionally purified prior to evaporation in a Silica Gel 60 column (0.063–0.2 mm, Merck, Darmstadt, Germany).

### 4.7. AFB1 Quantification in Buffer Solutions and AFO-Exposed Grain 

The content of residual AFB1 in model buffer solutions and grain extracts was measured via HPLC using a Waters 1525 Breeze chromatograph (Waters Corp., Milford, MA, USA). An aliquot (10 µL) of each tested sample was applied in a Symmetry C18 temperature-controlled (27 °C) column (5 µm, 150 × 4.6 mm, Waters Corp., Milford, MA, USA). AFB1 was eluted with a 0.3% H_3_PO_4_:acetonitrile (1:1) mix and detected at 362 nm using a Waters 2487 UV detector (Waters Corp., Milford, MA, USA). The aforementioned commercial AFB1 was used as a reference sample (Appendix A). The AFB1 content was calculated using a calibration curve plotted for the reference sample in the zone of a linear dependence of a peak area on the AFB1 amount. All samples were analyzed in triplicate. AFB1 reduction in contaminated samples was expressed as a percentage of the control (inoculated grain, which was not exposed to AFO). The toxin recovery level determined in experiments with the addition of reference AFB1 to the non-inoculated grain samples prior to their extraction reached 77 (wheat) and 75% (corn). The limit of detection for the grain extract was 0.005 µg/mL. All measurements were carried out in at least three analytical replicates.

### 4.8. Statistical Treatment 

To confirm the significance of difference between the mean values of treated and control variants as well as between means of variants involving enzymatic treatments, the *t*-test for independent variables was used (STATISTICA v. 6.1 software package; StatSoft Inc., Tulsa, OK, USA). The difference was considered to be significant at *p* ≤ 0.05. 

## Figures and Tables

**Figure 1 toxins-15-00678-f001:**
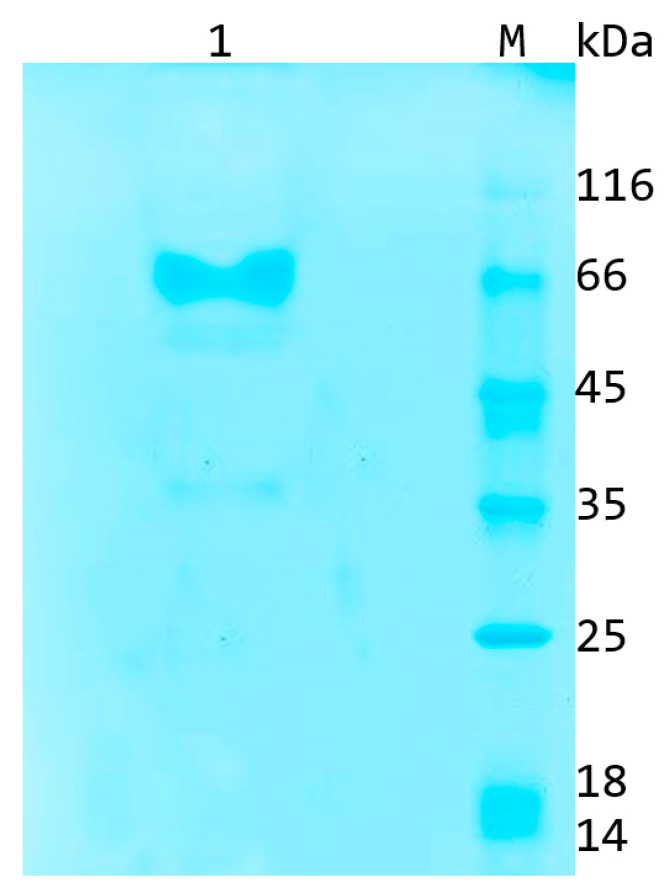
SDS-PAGE electrophoregram of a 72 kDa recombinant AFO (1) purified from the cultural liquid of *P. pastoris* GS115 by means of immobilized metal affinity chromatography. M, molecular weight marker.

**Figure 2 toxins-15-00678-f002:**
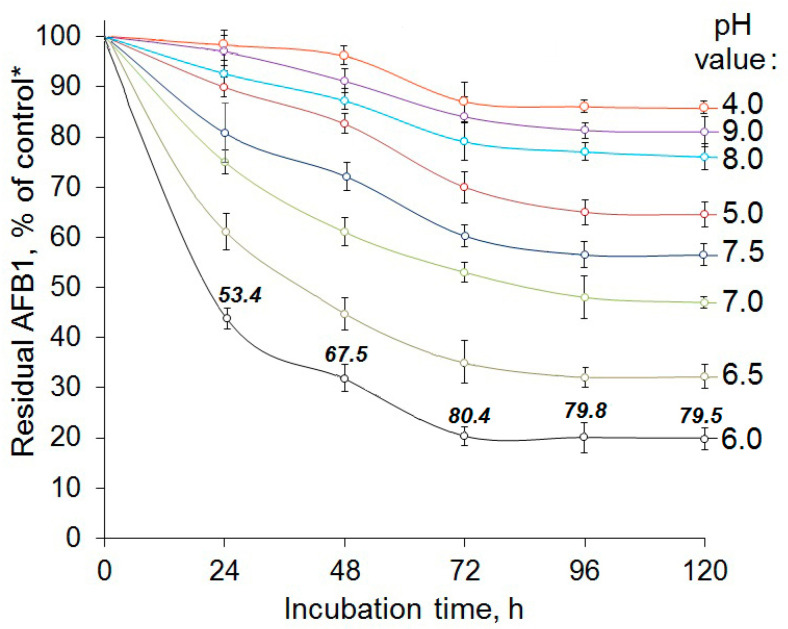
The effect of pH on the dynamics of AFB1 removal from model solutions exposed to recombinant aflatoxin oxidase (AFO) expressed in *Picha pastoris.* Numbers above the curve points indicate the average AFB1 reduction (%) in a buffer solution after exposure to AFO. All results were calculated as the mean of three experiments ± standard deviation (SD). * Here and in Figure 3: the AFB1 content detected by HPLC in the corresponding AFO-free buffer solution during a 120 h incubation was considered as a control.

**Figure 3 toxins-15-00678-f003:**
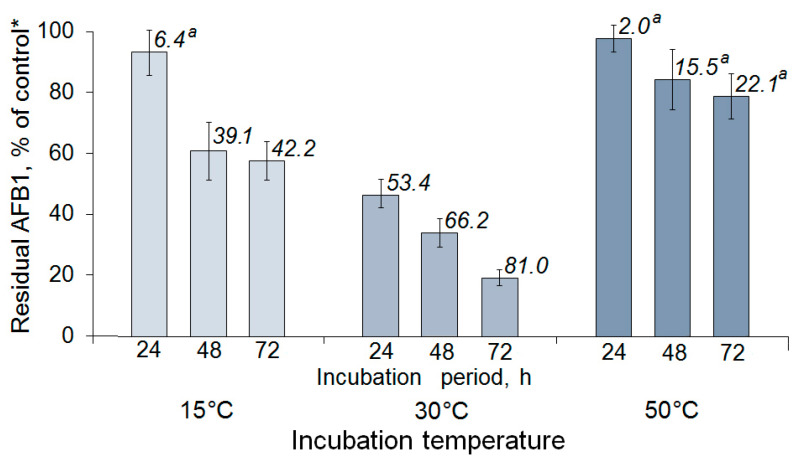
Effect of different temperatures on the rate of AFB1 removal from model solutions by recombinant aflatoxin oxidase (AFO). Results are presented as the mean of three experiments, each arranged into three replicates. Error bars indicate SD. Numbers above the columns show AFB1 degradation (%). Insignificant difference between the control and enzymatic degradation is indicated with “a”. * See explanation of the control in Figure 2.

**Figure 4 toxins-15-00678-f004:**
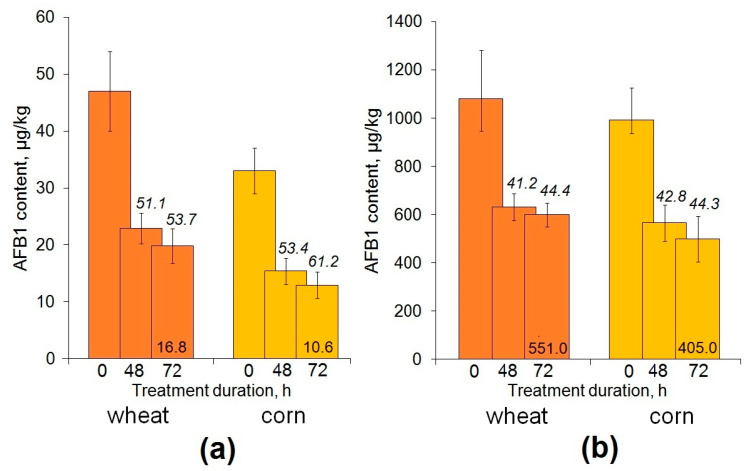
Enzymatic reduction in the AFB1 contamination level in wheat and corn grain artificially inoculated with a toxigenic *A. flavus* strain and incubated for 3 (**a**) and 7 (**b**) days. Numbers above the columns (in italic) indicate the average percent of the toxin degradation; numbers at the column bottoms show the minimum level of residual AFB1 (µg/kg) that was achieved after a 72 h treatment of grain with recombinant AFO. The difference between the control (non-exposed to the enzyme, 0 h) and the treated grain samples was significant at *p* ≤ 0.05.

## Data Availability

Data are contained within the article and Appendix A.

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
