# Peer review of "Recombinant Oxidase from Armillaria tabescens as a Potential Tool for Aflatoxin B1 Degradation in Contaminated Cereal Grain"

_toxins, 2023, doi:10.3390/toxins15120678_

Round 1

Reviewer 1 Report

Comments and Suggestions for Authors

According to the Turnitin analysis results, the similarity reaches 38%. Especially in the materials and methods writing content (sections 5.1, 5.7, 5.8), 6% is related to this article https://www.mdpi.com/2072-6651/12/8/475. If it is a self-citation, please rewrite it.

Please see the pdf file for review comments.

Comments on the Quality of English Language

The quality of the English language in the manuscript is good. Overall, the manuscript reads well, and I have only minor suggestions for improving clarity and flow in a few places. The authors have a good command of scientific English. With the minor edits, this manuscript would be suitable for publication.

Author Response

  1. According to the Turnitin analysis results, the similarity reaches 38%. Especially in the materials and methods writing content (sections 5.1, 5.7, 5.8), 6% is related to this article https://www.mdpi.com/2072-6651/12/8/475. If it is a self-citation, please rewrite it.

RESP.1. We have tried to re-write the aforementioned sections to reduce the content similarity with one of our earlier article. Note that the submitted manuscript is one of our regular publications describing obtaining and testing recombinant mycotoxin-degrading enzymes. Some key approaches, experimental design, and methods are similar or only in detail vary depending on the toxin, enzyme and its microbial producers. As we found, the Turnitin can reacted to some common phrases and word sequences such as “the AFB1 removal from model solutions”, “grain contamination/decontamonation”, “degradation activity after incubation in model solutions”, “monitoring of feed contamination with AFB1” etc.

    Nevertheless, we tried to correct the text where possible, but some descriptions, especially in “Introduction” and "Materials and methods" can not be changed without worsening of the language quality or description accuracy. The deeper alterations can looks confusing for experts in the field and cause misunderstanding. We hope the performed changes will be enough.

  1. what is a very high sensitivity of farm poultry?

RESP.2. For example, LD50 for ducks is only 0.3-0.6 mg/kg of body weight. We added LD50 of AFB1 for some poultry species to illustrate this phrase.

  1. Elaborate on prior research characterizing A. tabescens oxidase and its efficacy in degrading other mycotoxins. Explain why this oxidase is a promising enzyme for aflatoxin degradation based on its catalytic mechanism, substrate specificity, etc.

Summarize key prior studies that used oxidoreductase enzymes for aflatoxin decontamination. Note the enzymes used, their limitations, and how the current work builds upon the existing work

Thank you very much for your suggestions for improving the manuscript. We followed your recommendations. All changes are indicated in yellow.

     RESP.3. Taking into account your comments, we added data on the prior research of AFO from A. tabescens [Liu et al., 2001; Cao et al., 2011; Wu et al. 2015; Sinelnikov et al., 2022] and its ability to degrade another AFB1 precursor, versicolorin A, specified the reactive site and the Km value of the enzyme, and highlighted the role of hydrogen peroxide produced during the enzymatic reaction in a detoxification of AFB1-epoxide [Wu et al., 2015].

     We also briefly summarized and added information related to AFB1-degrading oxidoreductases from xylotrophic basidiomycetes and mentioned the existing limitations for manganese-dependent peroxidase of Phanerochaete sordida  and laccases from Trametes versicolor, Pleurotus ostreatus, Peniophora sp. as well as laccase secreted by recombinant A. niger (see please the highlighted text and new references 35-41 in the updated manuscript).

    Some other explanations, why AFO from A. tabescens is a promising enzyme for the AFB1 degradation, have already been given in the manuscript in the Introduction on lines 51-58 (mentioning of AFO properties based on its mode of catalytic action [Cao et al., 2011; Wu et al., 2015], a high affinity to AFB1 [Cao et al., 2011], reduction of the toxicity and mutagenicity of AFB1 [Liu et al., 2001], and the ability of the enzyme to destroy not only this mycotoxin, but also sterigmatocystin [Wu et al., 2015] ). In addition, other AFO properties facilitating possibility of its heterologous expression (knowledge about crystal structure of this monomeric enzyme and the sequence of the encoding gene [Xu et al., 2017]) as well as information on the safety of products resulted from the enzymatic AFB1 conversion [Guan et al., 2921] were indicated in the Discussion. We would prefer to remain these data in Discussion to avoid breaking of the context logic of manuscript and an excess length of the Introduction.

  1. what is the positive  control for the A. flavus grain inoculations

RESP.4. The positive control in experiments involving wheat and corn grain treatments with AFO was grain inoculated with the AFB1-producing A. flavus strain А24 (deposited as a highly toxigenic strain at the State Collection of Plant Pathogenic Microorganisms (SCPPM) of the All-Russian Research Institute of Phytopathology) and treated with 20 mM AFO-free phosphate buffer, pH 6.0. This strain was received from the SCPPM and stored at our working collection. This positive control was described in the text as the inoculated control, while non-inoculated grain was indicated as “non-inoculated control” and served as the negative control. We expanded the corresponding explanations in subsections 5.5 and 5.6.

  1. The rationale for the specific AFO incubation conditions (Figs.2-3) is not provided. These factors should be mention in introduction

RESP.5. We included the corresponding rationale in the subsection 2.2. and briefly mentioned this matter in the Introduction

  1. The section does not mention any controls or replicates used in the study. It is crucial to include this information to ensure the validity and reliability of the experimental results. Please provide details regarding the control group and the number of replicates used for each experiment.

RESP.6. Controls used in the experiments on the AFB1 removal from the model solutions were described in the subsection 5.4 of “Material and Methods” (The corresponding buffers supplemented with the toxin at the same concentration were used as controls.) and were also mentioned in figure captions for Figs. 2 and 3 (*The content of AFB1 detected by HPLC in the corresponding AFO-free buffer solution during a 120-h incubation was considered as the control). Caption of Fig. 3 mistakenly referred a reader to Fig. 1 instead Fig. 2. This is corrected now.

Controls for grain decontamination experiments have been re-indicated in the corresponding subsections of the “Materials and Methods” (see, please, the above response to comment # 4).

Information about the number of biological or technical replications for each experiment was included in each corresponding subsection of the Materials and Methods section.

  1. It is unclear how the materials used in the study were selected. Please clarify the criteria for choosing the specific materials and provide a rationale for their inclusion in the experiment.

RESP.7. In this study we mainly used standard strains and reagents applied for heterologous expression. Due to a small number of reagents and strains used, we did not put their description in a separate section.  

- P. pastoris strain GS115 is a standard commercial yeast strain commonly used in yeast experiments.

- YPD medium is a standard medium for yeast cultivation. 

- E. coli XLI-Blue is a standard strain used for the plasmid vector development.

- Primers were selected according to the sequence of the target gene.

- pPIGa-1 vector was successfully used in our earlier study for the expression of recombinant proteins in Pichia pastoris. The reference to the previous study was added into the Subsection 5.1.

- Zeocin was used, since the pPIG vector has a selective zeocin (ZeoR) resistance marker similar to the commercial pPICZaA vector; this is the standard choice for such studies.

- The culture medium for fermentation of the producer strain is included into the laboratory's own protocol representing a deeply modified protocol from Ghosalkar et al., 2008 (ref. 69, was added to the manuscript).

- Information about toxigenic A. flavus strain was included into the Subsection 5.5.

All changes and correction are highlighted in yellow.

Reviewer 2 Report

Comments and Suggestions for Authors

In the manuscript titled, “Recombinant oxidase from Armillaria tabescens as a potential tool for aflatoxin B1 degradation in contaminated cereal grain”, the authors recombinantly expressed a previously identified aflatoxin oxidase (AFO) from Armillaria tabescens in Pichia pastoris. The recombinant AFO showed activity against purified aflatoxin B1 as well as in grains (wheat and corns) contaminated with aflatoxin B1-producing Aspergillus flavus strain. The optimal pH and temperature of the enzyme were also determined.

I found the manuscript to be scientifically sound and interesting. Since this is not the first time that this enzyme was recombinantly expressed, the authors should address this question more thoroughly than already stated in the discussion. Also, if possible should compare AFO activity against other AFB1-degrading enzymes?

The main drawback of this manuscript is in English usage/structure. The manuscript should have been proofread by a native speaker.

Other minor details:

Line 30, “gain”, should be “grains”.

Line 75, Since the authors are using P. pastoris expression system, it is better to specify the system used instead of the general term, “Eukaryotic”.

Line 85, need proper reference.

Line 99 (Figure 1, legend): “cultural liquor”?

Line 327, Aspergillus flaus is a typo.

Comments on the Quality of English Language

In the manuscript titled, “Recombinant oxidase from Armillaria tabescens as a potential tool for aflatoxin B1 degradation in contaminated cereal grain”, the authors recombinantly expressed a previously identified aflatoxin oxidase (AFO) from Armillaria tabescens in Pichia pastoris. The recombinant AFO showed activity against purified aflatoxin B1 as well as in grains (wheat and corns) contaminated with aflatoxin B1-producing Aspergillus flavus strain. The optimal pH and temperature of the enzyme were also determined.

I found the manuscript to be scientifically sound and interesting. Since this is not the first time that this enzyme was recombinantly expressed, the authors should address this question more thoroughly than already stated in the discussion. Also, if possible should compare AFO activity against other AFB1-degrading enzymes?

The main drawback of this manuscript is in English usage/structure. The manuscript should have been proofread by a native speaker.

Other minor details:

Line 30, “gain”, should be “grains”.

Line 75, Since the authors are using P. pastoris expression system, it is better to specify the system used instead of the general term, “Eukaryotic”.

Line 85, need proper reference.

Line 99 (Figure 1, legend): “cultural liquor”?

Line 327, Aspergillus flaus is a typo.

Author Response

I found the manuscript to be scientifically sound and interesting.

  1. Since this is not the first time that this enzyme was recombinantly expressed, the authors should address this question more thoroughly than already stated in the discussion. Also, if possible should compare AFO activity against other AFB1-degrading enzymes?

Thank you for your positive appraisal of our work.

RESP.1. We have included additional information about recombinant AFO in the Discussion and Introduction sections, focusing mainly on the enzymes that were expressed in P. pastoris and used for grain decontamination, and indicated their degradation effect. We planned to compare AFO activity with other AFB1-degrading enzymes when preparing the first version of the manuscript, but faced with a number of complications. A correct comparison of the target activity is hindered to a great extent by incomplete or incomparable data published. For example, concentrations of the affected toxin or the quality of the enzyme are widely varying; they are not always indicated or shown either in different ways (final content per mL of incubation media, initial molarity, etc.), or using different methods of AFB1 quantification. As to grain decontamination, other AFOs expressed in P. pastoris were either isolated from another fungus (Trametes versicolor) and applied on naturally polluted peanut (AFO from T. versicolor), and the initial infection level was not detected, or on mold corn (AFO from A. tabescens) with the lower АFB1 contamination level, where dynamics of the decontamination was not presented, and AFB1 degradation percent was determined by ELISA, not HPLC.

  1. The main drawback of this manuscript is in English usage/structure. The manuscript should have been proofread by a native speaker.

RESP.2. The manuscript language was edited by experienced colleague, who has already prepared a number of papers for MDPI, which were published with only a minor language editing by the MDPI staff.

  1. Other minor details:

Line 30, “gain”, should be “grains”.

CORRECTED

Line 75, Since the authors are using P. pastoris expression system, it is better to specify the system used instead of the general term, “Eukaryotic”.

DONE

Line 85, need proper reference.

DONE

Line 99 (Figure 1, legend): “cultural liquor”?

CORRECTED

Line 327, Aspergillus flaus is a typo.

DONE

The manuscript text was checked, other typos found were corrected.

All changes and corrections excepting minor language editing are highlighted in yellow.